# Machine learning analysis of volatolomic profiles in breath can identify non-invasive biomarkers of liver disease: A pilot study

**Jonathan N. Thomas, Joanna Roopkumar, Tushar Patel** [ID]*

Department of Transplantation, Division of Gastroenterology and Hepatology, Mayo Clinic, Jacksonville, Florida, United States of America

* patel.tushar@mayo.edu

## Abstract

Disease-related effects on hepatic metabolism can alter the composition of chemicals in the circulation and subsequently in breath. The presence of disease related alterations in exhaled volatile organic compounds could therefore provide a basis for non-invasive biomarkers of hepatic disease. This study examined the feasibility of using global volatolomic profiles from breath analysis in combination with supervised machine learning to develop signature pattern-based biomarkers for cirrhosis. Breath samples were analyzed using thermal desorption-gas chromatography-field asymmetric ion mobility spectroscopy to generate breathomic profiles. A standardized collection protocol and analysis pipeline was used to collect samples from 35 persons with cirrhosis, 4 with non-cirrhotic portal hypertension, and 11 healthy participants. Molecular features of interest were identified to determine their ability to classify cirrhosis or portal hypertension. A molecular feature score was derived that increased with the stage of cirrhosis and had an AUC of 0.78 for detection. Chromatographic breath profiles were utilized to generate machine learning-based classifiers. Algorithmic models could discriminate presence or stage of cirrhosis with a sensitivity of 88–92% and specificity of 75%. These results demonstrate the feasibility of volatolomic profiling to classify clinical phenotypes using global breath output. These studies will pave the way for the development of non-invasive biomarkers of liver disease based on volatolomic signatures found in breath.

## Introduction

The liver has a central role in metabolism, and disease related effects on hepatic functioning can alter the nature and quantity of metabolites that are generated. Amongst these are volatile organic compounds (VOC), high vapor pressure molecules that can diffuse through the circulation and eventually be exhaled in the breath. While VOC only account for <1% of breath, hundreds of high vapor pressure molecules associated with systemic metabolic functions can be detected within each breath [1]. Thus, alterations in the VOC metabolomic (volatolomic) output of the liver associated with disease pathophysiology can be detected in exhaled breath

meet the criteria for access to confidential data following request and approval by the Mayo Clinic IRB (contact phone 507-266-4000), 201 Building 4-60, 200 First Street SW, Rochester, MN 55905.

**Funding:** Support for this study was provided by the Mayo Clinic, and the James C and Sarah K Kennedy Deanship and the Alfred D. and Audrey M. Petersen Professorship of Cancer Research (TP) The funders had no role in the study design, data collection and analysis, decision to publish, or preparation of the manuscript.

**Competing interests:** The authors have declared that no competing interests exist.

[2]. This phenomenon has been recognized for millennia, forming the basis of *fetor hepaticus* and other breath-based manifestations of disease.

The application of VOC analysis to capture disease relevant information from exhaled breath provides an untapped opportunity to develop non-invasive biomarkers for liver diseases that may facilitate earlier diagnosis or guide patient management. Chronic liver disease and cirrhosis may be present in the absence of symptoms but yet are a major cause of morbidity and mortality [3, 4]. A timely diagnosis of cirrhosis may enable interventions to limit inflammation or progression of fibrosis as well as the initiation of surveillance approaches for early detection of hepatocellular cancer. Once cirrhosis is present, decompensation is clinically defined by the onset of complications such as ascites, and portends a higher risk of morbidity, hospitalizations, prolonged care and mortality [5]. Furthermore, the hepatic volatolomic output could potentially be altered as a consequence of progressive portal hypertension and hepatic dysfunction prior to the onset of clinical manifestations of decompensation.

Although prior studies have described and analyzed VOC in the breath of patients with liver diseases, the feasibility of using breath VOC analysis for disease detection remains poorly defined. Accurate identification of individual VOC is highly dependent on the detection technology used. This has varied across studies and has confounded efforts to identify or catalog disease specific compounds. Consequently, there is a lack of consensus on the optimal use of individual or groups of VOC to differentiate between different clinical states. This has hampered the use of exhaled breath analysis for biomarker applications. Compared with the study of single VOC, global or broad volatolomic analysis would incorporate changes that are reflected within a wider range of low abundance disease associated VOC present in exhaled breath [6–9]. We performed a proof of concept study to establish the utility of breath volatolomic profiling to develop predictive models. A highly sensitive separation and detection approach to generate volatolomic profiles from exhaled breath samples was developed by combining thermal desorption (TD) with both gas chromatography (GC) and field asymmetric ion mobility spectrometry (FAIMS). This enabled us to capture multi-dimensional volatolomic data based on both chromatographic separation and ion-mobility spectrometry to separate ions based on their drift in high electric fields. The data was then combined with supervised machine learning to generate breath volatolomic based classifiers for the presence or stage of cirrhosis, thereby demonstrating the feasibility of using this approach to develop non-invasive biomarkers associated with liver diseases.

## Methods

### Ethical approval

The study was conducted under a Mayo Clinic Insitutional Review Board (IRB) approved protocol and conformed to the ethical guidelines of the Declaration of Helsinki. Informed consent was obtained from each participant in writing. The trial was registered at clinical trials.gov (NCT04341012).

### Study design and participants

The study was a prospective, single-institution study. All study participants were enrolled between September 2019 and March 2020. The study inclusion criteria were the ability to provide informed consent and age greater than 18 years. There were no exclusion criteria. Participants were categorized into groups based on absence or presence of cirrhosis and/or portal hypertension, or their complications as determined on the basis of histologic, clinical, biochemical or elastographic features. Participants with no cirrhosis or portal hypertension were designated as Stage 0. Participants with cirrhosis or portal hypertension were designated as

Stage 1, 2 or 3 based on the absence or presence of complications of portal hypertension (ascites, variceal hemorrhage, hepatic encephalopathy) or liver insufficiency (jaundice). Stage 1 had no varices or other clinically evident complications, Stage 2 had varices only but no other complications, while Stage 3 had decompensated disease, manifest with ascites, variceal hemorrhage, or hepatic encephalopathy. The clinical diagnoses were made independently by two hepatologists. All participants completed a questionnaire at the time of the breath collection regarding their lifestyle, recent dietary choices, current symptoms and other clinical information.

## Breath sample collection

Breath samples were collected using the ReCIVA breath sampler (Owlstone Medical, Cambridge, UK) and analyzed by TD-GC-FAIMS (Fig 1). Subjects were asked to fast for at least four hours prior to the breath collection, avoiding solid food prior to the collection. A breath sample was collected by a trained researcher using the breath sampler. Sample collection was performed with the patient seated upright, resting for at least 10 minutes. An air supply unit attached to the sampler pumps provided filtered, ambient air with reduced VOC at 40 L/min for the patient to inhale. 1 L of exhaled breath from both the upper and lower airways was collected, at a flow rate of 200 mL min-1, onto four preconditioned Bio-Monitoring TD tubes (Markes International, South Wales, United Kingdom). The breath sampler uses pressure and $CO_2$ sensors to monitor the patient's breathing rate to regulate the timing of its two pumps for the four sorbent tube ports. This allowed for control over the total volume, flow rate, and the specific phase of exhaled air collected.

## Collection of environmental sample blanks

Room air sample and air supply collections were performed immediately after the breath sample collection using the ReCIVA breath sampler and TD tubes from the same conditioned batch as the breath samples. For room air sample collection, the breath sampler was placed sideways on a pre-cleaned metal surface, facing outwards, with the air supply on. The ReCIVA

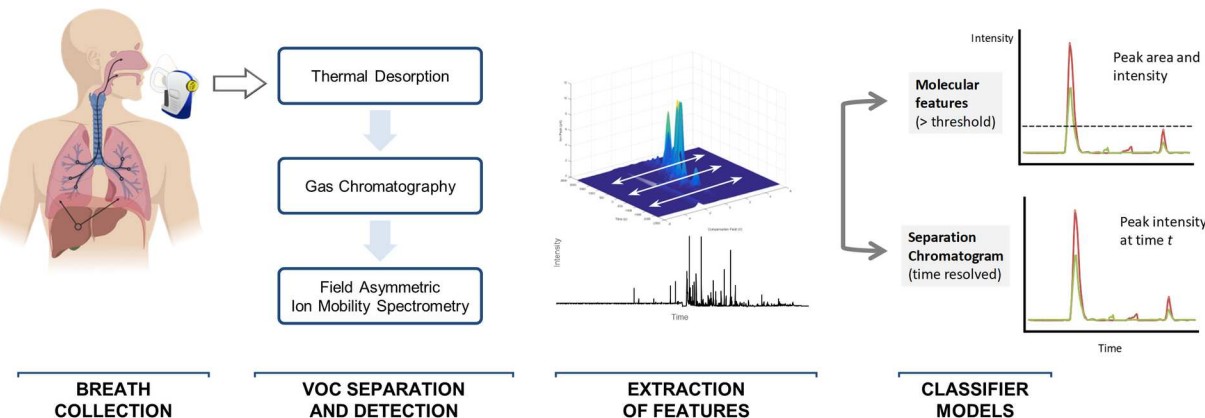

**Fig 1. Collection and analysis of breath samples.** Volatile organic compounds (VOC) in exhaled breath are collected onto thermal desorption tubes that pre-concentrate and focus analytes into a gas chromatography (GC) column for physical separation. Subsequently, VOC are further distinguished by field asymmetric ion mobility spectrometry that applies an alternating electric dispersion field with maximum voltage of 45V, 55V or 65V. Data output matrices for both positive and negative ions detected are preprocessed and the maximum ion peak intensity identified for each positive and negative ion resolved by GC retention time. Molecular features describe intensity resolved VOC defined by intensity greater than a threshold level, whereas separation chromatograms capture all VOC in a time-resolved manner. Differences in VOC output measured as molecular features or in separation chromatograms between controls (green lines) and disease states (red lines) are analyzed by machine learning to generate classifier models. Image generated with *Biorender.com*.

was set to keep one pump (right) always on and collect 1 L of room air at 200 mL min-1 using either one or two sorbent tubes. During air supply collection, the ReCIVA and field blank collection tubes were strapped securely to a pre-cleaned glass head and set to collect onto the other one or two tubes (left) using the same parameters. The cart and glass head were cleaned using a 70% ethanol solution or isopropyl wipes at least 1 hour prior. Environmental blanks were stored, transported, and analyzed alongside corresponding breath samples.

## Preconditioning of thermal desorption tubes

Prior to breath sample collection, TD tubes were preconditioned using a TC-20 TD device (Markes International, South Wales, UK) with 55 to 60 mL min$^{-1}$ nitrogen (99.9999%) gas flow at 330°C for at least 2 hours. Tubes were capped with stainless steel travel caps with Viton O-rings (Owlstone Medical, Cambridge, UK) if the collection was within 7 days; or with brass caps fitted with polytetrafluoroethylene ferrules if stored for a longer period. All tubes were wrapped in non-coated aluminum and were placed in aluminum screw-top canisters, sealed with aluminum wrap, and stored at 4°C. Further, the wrapped tube canisters were transported via a cooler to the clinic before and after collection. These additional measures were taken to help prevent the contamination, loss of sample, and slow diffusion of analytes across the different sorbent beds inside the tube, a porous polymer and graphitized carbon [10].

## Separation and isolation of VOC using TD-GC-FAIMS

Thermal desorption was carried out using a Unity-xr TD unit (Markes International, South Wales, UK) equipped with a material emissions cold trap. During analysis, each tube was pre-purged with nitrogen gas for 10 minutes with split flow on at 50 mL min$^{-1}$. Sample tube desorption was set to 120°C for 10 minutes at 50 mL min$^{-1}$ onto the 0°C cold trap solely. The trap was purged for 2 minutes, then heated at a rate of 100°C min$^{-1}$ to 140°C and held for 6 minutes, sending VOC through the 130°C TD-GC transfer line. Separation was performed using a Trace 1310 GC (Thermo Fisher Scientific, Waltham, Massachusetts) coupled with a Lonestar FAIMS detector (Owlstone Medical, Cambridge, UK). VOC were separated on a HP-5 (Agilent, Santa Clara, California) fused silica GC column (30m length × 0.25 μm thickness × 0.32 mm inner diameter) with helium (99.999%) carrier gas. GC controls were set for an initial 40°C hold of 2 min, ramp to 120°C at 5°C/min, hold for 2 min, and final ramp to 200°C at 8°C/min for a final hold of 6 min. Medical-grade clean air was introduced into the 130°C GC-FAIMS transfer line at a rate of 2700 mL min-1, providing the reactive ion cloud needed for ionization of emerging analytes.The FAIMS was configured such that the magnitude of the alternating electric field, or dispersion field (DF), voltage cycled through 45 V, 55 V, and 65 V with a total of 5192 scans across the GC runtime. The compensation field (CF) voltage scanned to correct differential ion drift across 512 preset increments between– 6 V and 6 V direct current. Detected ion intensity was measured over a range from 0 to 10 pA.

## FAIMS data processing

Data from FAIMS were pre-processed and parsed using an automated MATLAB 2019b (MathWorks, Natick, MA) processing pipeline to (1) separate the ion intensity matrices for each DF setting, (2) subtract out environmental VOC and background current fluctuations using room air or air filter field control blanks (as required), and (3) generate a max peak ion chromatogram for each sample. The first step involved parsing the raw DF settings, and then combining the negative and positive ion intensity matrices to generate three FAIMS DF-specific matrices (512 CF scan points by 3460 time charge points). Next, environmental samples were directly subtracted from their corresponding breath samples across the entire 1.77

million data points to generate a separate dataset for later classification analysis. To further simplify the data to a single axis, the outer matrix cells with values below the overall max base-line intensity (0.0104 pA) were eliminated, limiting the matrix to 256 CF scans between -3 V and 3 V and removing the terminal ~40 seconds (60 time charge points) of the GC run The maximum intensity value across all CF scans for each time resolved point was selected, simplifying the ion peaks to a single time charge axis S1 Fig. The breath sample data were now represented by the resulting three DF-specific time resolved separation-based chromatograms (SC), comprised of 3400 ion intensity values S2 Fig.

## Supervised machine learning analysis

For generating disease state classifiers, SC were imported into MATLAB 2019b Classification Learner App.. A training set comprised of six randomly selected patients from each of stages 0, 1, 2 and 3. Twenty-four classifier model types were trained and subsequently tested for each analysis, generating a confusion matrix and model performance characteristics using 5-fold cross-validation. The classifiers with the best performance metrics (AUC > 0.7) were selected and then further tested and evaluated in an independent external validation set.

# Results

## Intra-individual variability of volatolomic detection

To analyze intra-individual variability, breath samples were obtained from a single healthy individual and using a standard protocol. Five breath samples comprising of one liter of breath were collected on five separate days within a 7-day period. Each sample was collected after an overnight fast of at least six hours with only water, and collected between 7:30 and 10:00 AM. Data was collected using FAIMS DF settings of 45 V, 55 V, or 65 V, at a CF of 0.55V. From each dataset, molecular features (MF) along the positive reactive ions detected were identified and analyzed. The max positive ion peak intensity derived from the background air passed through the FAIMS was 0.391 pA. MF were thus defined as those with distinctive retention times and peak maximum intensity beyond a threshold set at 0.5 pA. MF may reflect one VOC, although superposition of peaks can occur in some instances due to co-elution. Of note, even small shifts in ion intensity could reflect the presence of individual low abundance VOC. The thresholds set therefore enabled elimination of all background fluctuations, and focused the analysis on the most abundant volatolomic content in the sample. First we analyzed the variability in MF detected in technical replicates collected on the same day. The overall coefficient of variation (CV) in technical replicates across all DF settings was 5.7%. Next, we determined the biological variation in detection of MF from day-to-day. An average of 60.1 MF were detected across all settings from day-to-day, with a CV of 14.8%. Next, we evaluated the detection of MF and variability at different DF settings. The average number of features varied at each DF setting, with far fewer detected at DF 65 V when compared with either DF 45 V or DF 55 V (Fig 2). The overall CV in number of MF detected ranged from 13% at DF 55 V to 34% at DF 65 V.

## Study subjects

The study population comprised of 50 subjects, with an equal proportion of males and females. 86% of study participants were white and 90% were non-Hispanic. The mean age of the population was 55.4 years and the overall mean BMI was 29.8. All except one individual were non-smokers. None of the participants reported any known occupational exposure to vapors. None of the participants reported any kind of upper respiratory infection. Other self-reported

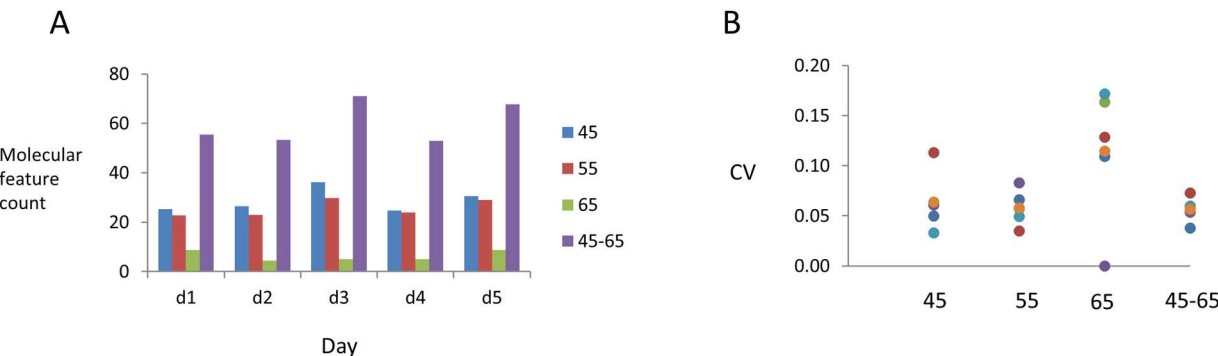

**Fig 2. Intra-individual variability of molecular features detected in breath analysis.** Molecular features (MF) were defined based on identified peaks with an intensity greater than a threshold of 0.5 pA. Twenty breath samples were collected using a standardized protocol from a single healthy volunteer over a five-day period. Chromatograms were extracted at a pre-defined compensation field at dispersion field (DF) settings of 45 V, 55 V or 65 V, and the number of MF present were quantitated. (A). The total number of MF identified at each DF setting, or with all three DF combined, on each of separate collection day. (B). The coefficient of variation (CV) of separate analyses at each DF setting alone or with all three DF combined is depicted. Colored dots indicate day of sample collection.

symptoms included cough (51%), dyspnea (18%), abdominal pain (38%), diarrhea (15%) and halitosis (11%) within the two weeks preceding sample collection.

All group and stage designations were verified by two experienced hepatologists with full consensus. 11 study participants did not have cirrhosis or portal hypertension and were designated as stage 0. Cirrhosis or portal hypertension was present in 39 participants. Of these the primary etiology was non-alcoholic steatohepatitis (21 persons), chronic hepatitis C virus infection (5 persons), alcohol-related liver disease (6 persons), hemochromatosis (1 person), primary sclerosing cholangitis (2 persons), and non-cirrhotic portal hypertension (4 persons). Of these 39 participants, 14 were designated as stage 1 (no ascites, no varices), 15 as stage 2 (with varices present but no ascites), and 10 as stage 3 (with decompensated disease). Two persons had hepatic encephalopathy. None of the participants had severe stage 4 disease as defined by a history of recurrent variceal hemorrhage, refractory ascites, and hyponatremia or hepatorenal syndrome. In these participants, the median Model for End Stage Liver Disease (MELD) score, was 10 with a range from 6 to 28, the median AST to platelet ratio index (APRI) was 0.744 with a range from 0.215 to 3.539, and the median Fibrosis-4 (FIB_4) score, was 3.37 with a range from 0.59 to 14.82. Detailed characteristics of the groups are described in Table 1.

## Sample collection and storage

A standardized set of instructions and protocol was used. A four-hour fast was required. However, most subjects had fasted overnight as collections were scheduled during the early morning. 350 samples (200 exhaled breath collections and 150 room air or air filter collections) were collected for this study. Samples were stored in sorbent tubes for a median of 7 days prior to TD-GC-FAIMS analysis. When long-term stored samples were removed for studies of optimized conditions, the median storage was 4 days.

## Analysis of molecular features

MF, bracketed within time defined parameters, were identified using the three DF-specific SC from each breath sample. The peak maximum ion intensity and peak area were calculated for each MF, which were averaged across all technical replicates for each patient. We examined the variability of each MF within biological replicates with age, or underlying etiology. A

**Table 1. Study subjects.**

| | Stage 0 | Stage 1 | Stage 2 | Stage 3 | Overall |
|---|---|---|---|---|---|
| | (n = 11) | (n = 14) | (n = 15) | (n = 10) | (n = 50) |
| Age mean (SD) | 43.8 (12.2) | 56.2 (10.4) | 58.1 (13.2) | 62.8 (11.0) | 55.4 (13.2) |
| Age median (range) | 45 (24–60) | 57.5 (35–69) | 61 (33–76) | 64.5 (42–74) | 57 (24–76) |
| n,% female | 6 (54%) | 6 (42%) | 8 (53%) | 5 (50%) | 25 (50%) |
| n, % white | 5 (45%) | 13 (92%) | 15 (100%) | 10 (100%) | 43 (86%) |
| n, % Hispanic | 1 (9%) | 1 (7%) | 3 (20%) | 0 | 5 (10%) |
| Cirrhosis (number) | 0 | 13 | 12 | 10 | 35 |
| Body mass index, mean (SD) | 29.2 (5.9) | 30.8 (5.6) | 29.7 (6.8) | 29.1 (5.2) | 29.8 (5.8) |
| n,% nose breathers | 10 (90%) | 10 (71%) | 10 (67%) | 6 (60%) | 36 (72%) |
| n,% cough | 0 | 1 (7%) | 2 (13%) | 2 (20%) | 5 (10%) |
| n, % shortness of breath | 0 | 1 (7%) | 3 (20%) | 1 (10%) | 5 (10%) |
| Mean duration of fast (hours) | | | | | |
| Solid foods | 12.8 | 13.5 | 11.7 | 10.4 | 12.2 |
| Liquids | 5.8 | 4.7 | 4.9 | 1.4 | 4.3 |
| Most recent use of live yoghurt (n, %) | | | | | |
| More than one week | 8 (72%) | 9 (64%) | 13 (86%) | 4 (40% | 34 (68%) |
| Within past week | 2 (18%) | 2 (14%) | 1 (7%) | 4 (40%) | 9 (18%) |
| Within past day | 1 (10%) | 3 (22%) | 1 (7%) | 2 (20%) | 7 (14%) |

greater variability was observed with the latter, particularly in samples from participants aged 45yr or younger S3 Fig. We first analyzed both peak max and area separately within each disease group to determine which was more informative. A one-tailed paired student's t-test was used to compare these between samples from healthy controls, participants with stage 0, and those with stage 1/2/3 disease. Ten MF with peak intensity and eight MF with peak areas were identified that had >30% differences between these two groups, with p values < 0.05. The area under a receiver operating characteristic curve (AUC) was determined for each one of these. The AUC ranged from 0.547 to 0.785 for individual MF peak intensity, and from 0.379 to 0.774 for individual MF peak area. There was a strong correlation ($R^2$ = 0.93) between AUC for peak intensity and AUC for peak area for individual MF. These findings indicated that the use of either the peak max intensity or the peak max area would suffice for analytical use to generate models. Eight unique features had an AUC greater than 0.7 (Fig 3). Amongst these, four showed a trend to increase with disease stage. We postulated that these MF would be more likely to reflect VOC that are directly impacted by liver function or portal hypertension. A logistic regression analysis of the MF with the highest AUC and associated with disease stage was performed and a *MF score* derived (Fig 4). Next, we assessed the relationship of the *MF score* to disease stage using MELD score and FIB-4 scores. The *MF score* was higher in Stage 3 disease than in Stage 1/2 disease. A MF score of 0.45 had a sensitivity of 90% and specificity of 57% for classifying the presence of cirrhosis or portal hypertension. The AUC of the *MF score* for classifying the presence of cirrhosis was 0.785. Thus, simple predictive scores can be generated from an analysis of intensity-threshold defined features in volatolomic analysis.

## Classifiers based on machine learning of global volatolomic output

Disease-associated alterations in VOC that have a low abundance may be detectable but may fall below arbitrarily defined intensity thresholds. To determine the utility of incorporating these minor, yet potentially informative volatolomic changes within a biomarker algorithm, we analyzed the entire GC-FAIMS output for each sample. Classifier models were generated

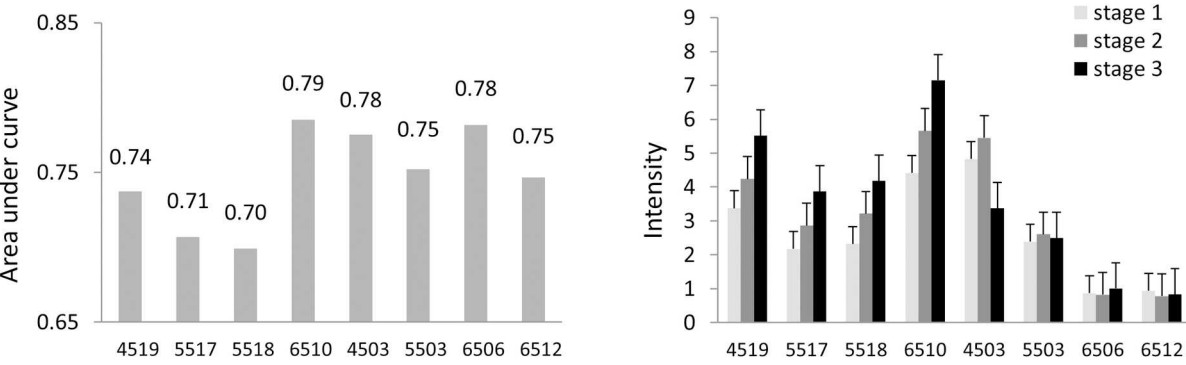

**Fig 3. Breath analysis of molecular features.** Breath samples were collected using a standard protocol from 50 patients. Chromatograms depicting the maximum intensity across all compensation fields at dispersion field (DF) settings of 45V, 55V or 65V were extracted. MF were defined based on identified peaks with an intensity greater than a threshold of 0.5 pA and designated with a four-digit number that included the DF setting as the first two, and order of separation as the second two digits. MF with differential ion intensities were isolated. The data represents the (A) the area under a receiver operator characteristic curve (AUC) for individual MF for the detection of cirrhosis; (B) MF peak intensity in samples from persons with different stages of disease. The data represents the mean and standard deviation for MF peak ion intensity in pA.

by using supervised machine learning in an unbiased approach to analyze the time resolved SC under DF 45 V. The average analytical run time was 2164.1 ± 1.6 (SD) seconds, with a range between 2160.2 to 2168.0 seconds. While minor, the inherent variability in run times (± 0.18%), can confound an assessment of VOC output.

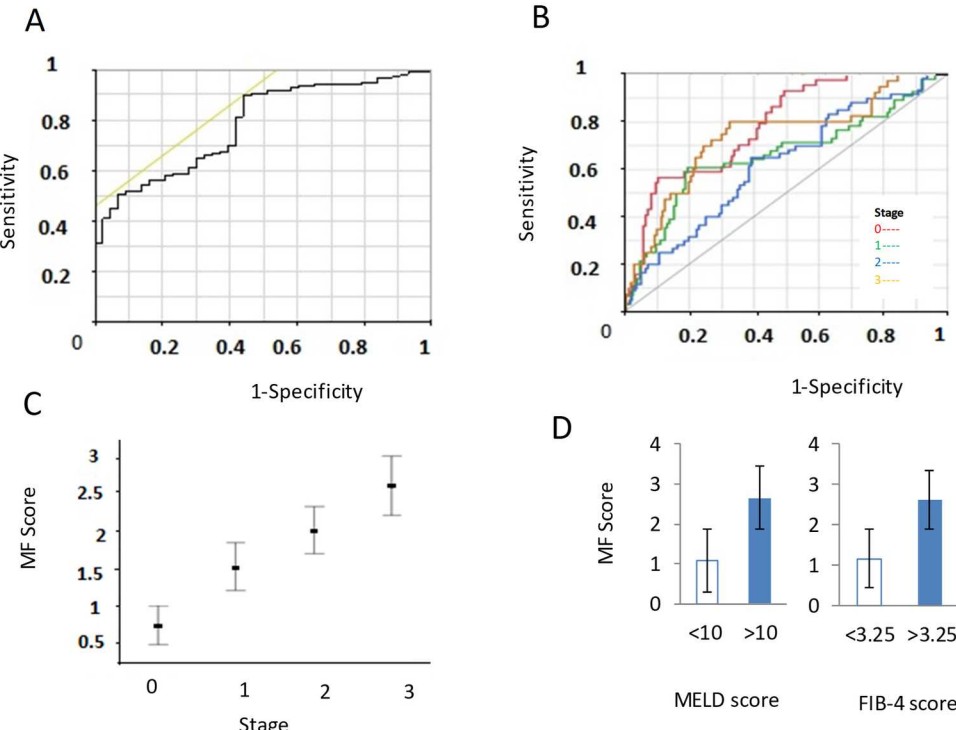

**Fig 4. Performance of _MF score_.** An _MF score_ was derived using logistic regression analysis for the molecular feature (MF) with the highest area under a receiver operating characteristic curve (AUC). Receiver operator curves for the _MF score_ for presence or absence of cirrhosis (A) or for the indicated cirrhosis disease stages (B). Variation in _MF score_ at different disease stages (C), and with Model for End Stage Liver Disease (MELD) or FIB-4 scores (D). The data represent mean and SD of the _MF score_.

To determine whether technical variations in volatolomic detection would preclude effective disease classification, we determined the inter-and intra-individual variability across different samples or participant technical replicates using a pre-trained convolutional neural network (CNN), ResNet-50. First, the entire FAIMS DF-specific matrices were imported into the fully connected CNN, generating 2048 intermediate prediction values that are used to determine the final categorization. We calculated the Euclidean mean distance (EMD) between these predictive values, providing a pair-wise measurement that reflects the dissimiliarity in ResNet-50 classification between samples. The average EMD across four biological replicates from a single healthy individual collected over five separate days was 1.44, and across technical replicates on each day was 1.39. In comparison, the average EMD for samples from a random selection of cirrhotic patients was 1.99 and in the healthy controls group was 2.45 S4 Fig. Thus, the variability across different individuals exceeded that occurring as a result of technical or biological variation within a single individual's FAIMS breath sample.

These data support the potential utility of global volatolomic analyses to develop classifier biomarkers. In order to further reduce technical variation, we eliminated samples that had been stored for more than 6 weeks prior to analysis, or where artifacts due to humidity contributed to signal degradation. 173 samples (87%) met these selection criteria. Machine-learning training and independent validation was done using MATLAB and the performance validated in an independent validation set.

Classifier model SC-2A was generated using ensemble learning using a random under-sampling boosted trees (RUSBT); it had a specificity of 75% with a sensitivity of 88% for the detection of cirrhosis (Fig 5). To evaluate the potential impact of environmental VOC on these classifiers, SCs were generated with the respective air supply filter blank sample subtracted or room air blank sample subtracted, and the impact on classifier model performance was assessed. Subtraction of either the air filter or room air data prior to model generation improved the sensitivity of the classifier models for detection of cirrhosis, with former being

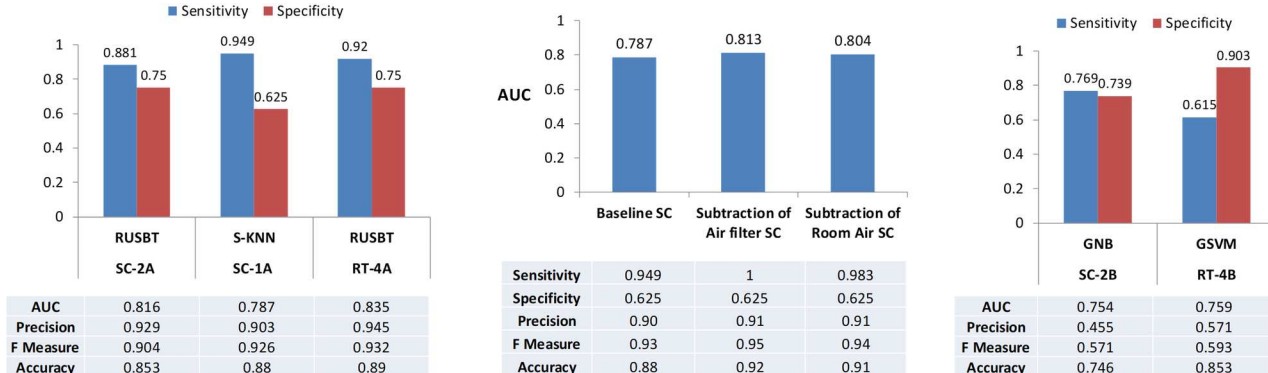

**Fig 5. Performance of volatolomic models for the detection of cirrhosis.** Volatolomic classifier algorithms were generated by machine learning based analysis of time resolved separation chromatograms (SC). (A.) Classifiers were generated for the detection of cirrhosis. Models were trained on a random set of samples from 24 patients, and the exported models' performance was assessed in an independent validation set of samples from the remaining patients. The sensitivity and specificity of models based on random under-sampling boosted trees (RUBST) or Subspace K nearest neighbors (SKNN) are shown along with performance characteristics for each model. (B.) Effect of environmental volatile compounds on performance of models for the detection of cirrhosis. Volatolomic classifiers were generated from analysis of baseline chromatograms, or after subtraction of concomitantly collected air-filter or room-air blank sample data. (C.) Classifiers were generated for the detection of decompensated disease (Stage 3) in persons with cirrhosis alone–Model SC-2B using Gaussian Naïve Bayes (GNB), or in the persons with cirrhosis or non-cirrhotic portal hypertension– Model RT-4B using Medium Gaussian support vector machines (GSVM). These models were trained on a random set of samples from 18 liver disease patients, and performance was assessed in an independent validation set of samples from 17 patients for SC-2B or 21 patients for RT-4B. The sensitivity and specificity of models are shown along with performance characteristics for each. AUC: area under the receiver operator characteristic curve.

100% sensitive. However, elimination of either room air or air filter flanks did not improve specificity for detection of cirrhosis.

While other models, such as SC-1A generated using Subspace $k$-Nearest Neighbors (SKNN) had a higher sensitivity of 94.9%, their specificity was lower. The performance of the models was similar across different stages S5 Fig. Notably, models trained on datasets that included cases of non-cirrhotic portal hypertension showed a higher sensitivity of 92% while maintaining specificity of 75%. Thus, changes related to portal hypertension may be important contributors to volatolomic outputs.

Additional models were generated for the detection of stage 3 disease in persons with known cirrhosis or portal hypertension. A classifier based on a Gaussian Naive Bayes (GNB) SC-2B had a specificity of 0.769 and a sensitivity of 0.739 with an AUC of 0.754. With the inclusion of data from patients with non-cirrhotic portal hypertension, classifiers for prediction of decompensated cirrhosis could be generated using Medium Gaussian support vector machines (GSVM) that had a higher specificity (0.903) albeit with a lower sensitivity. Preprocessing to subtract out room air blanks improved the sensitivity but not the specificity. However, the subtraction of air filter blank data did not improve either sensitivity or specificity.

Composite tandem models were generated by combining individual SC based classifiers for the prediction of cirrhosis that could also further separate into compensated or decompensated cirrhosis (Fig 6). Combining both RUSBT and GSVM classifiers into a single tandem model RT-4AB performed well in distinguishing either compensated or decompensated cirrhosis from those without cirrhosis. The tandem model had an accuracy of 89% for detection of the presence of cirrhosis, and 84% for the detection of decompensation when cirrhosis was present. A separate tandem model SC-2AB combining both RUSBT and GNB models and that included data from patients with non-cirrhotic portal hypertension had better performance with a sensitivity of 83% and specificity of 78% for detection of stage 3 disease. In conclusion, with particular attention to pre-analytical variables sample collection and processing, the use of automated machine learning derived models based on time resolved volatolomic profiling provide a higher performance alternative to the use of predictive scores based on intensity derived MF based scores in breath samples.

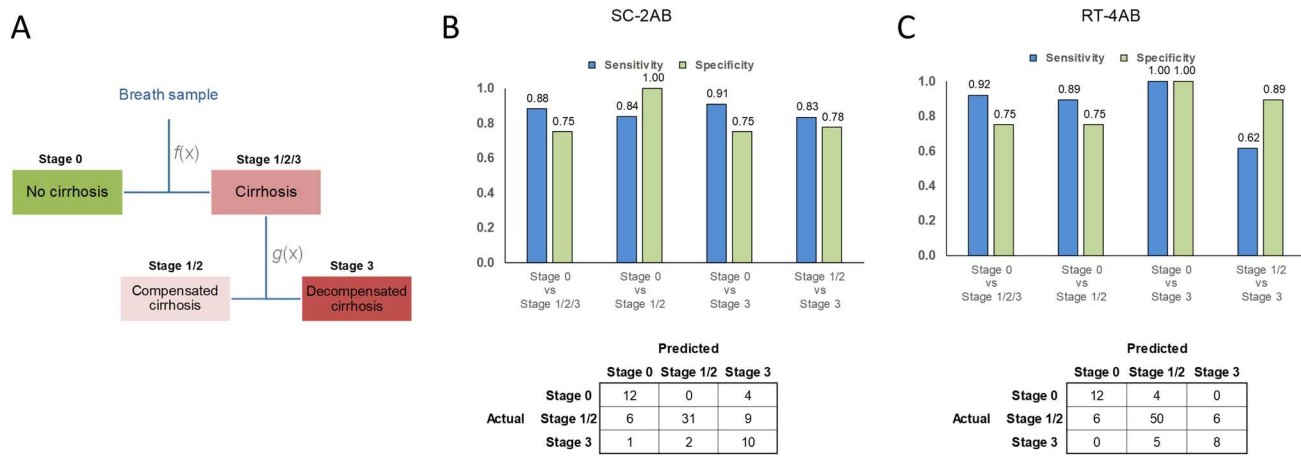

**Fig 6. Performance of tandem classifier models.** Tandem models were created by combining individual models for classification of cirrhosis and for classification of stage 3 (decompensated) disease. (A) In the tandem model, samples classified as cirrhosis using the former model would then be subsequently sub-classified into either compensated or decompensated disease using the latter. Models were trained and validated on a set of optimized samples, and performance validated on an independent set of samples using the exported tandem model. (B,C) The sensitivity and specificity (top) and confusion matrices (bottom) for tandem models for distinguishing between disease stages in independent validation cohorts are shown for subjects with (B) cirrhosis only (SC-2AB), or (C) on either cirrhosis or non-cirrhotic portal hypertension. (RT-4AB).

## Discussion

In this pilot study, we demonstrate the feasibility of a systematic approach to the detection of exhaled breath-based volatolomic profiles by illustrating their use for the detection of cirrhosis. These profiles capture the breadth of metabolomic activity without direct identification of individual VOC, and can capture information from low abundance VOC. The volatolomic profiles were generated by TD-GC-FAIMS as a three-dimensional data matrix comprising of time resolved ion intensities at different compensation field points. Intensity defined features derived from these data matrices can be used to generate a biomarker score whereas time-resolved features can be used to generate disease classifiers using machine learning. Thus both intensity and time resolved features of global breath volatolomic analyses could be used to generate clinically useful biomarkers that are distinctive, yet complementary. The multimodality separation approach combining GC for physical and time dependent separation with FAIMS for ion differential mobility separation provides higher resolution separation of VOC within a single work stream. Combining the data obtained with the experimentally derived algorithmic classifiers offers a platform that can be adopted within diagnostic laboratories.

The variability and sensitivity of VOC detection on breath analysis have limited the ability to develop breath-based biomarkers. Sources of variation can include environmental, technical, biological or patient-specific factors. Patient age, gender, diet, oral hygiene, smoking history, body mass, medical co-morbidities, and concomitant use of probiotics, antibiotics or other drugs could potentially impact on breath VOC changes. However in one study, alterations in hematological or biochemical markers such as white-blood cell count, cholesterol, or triglyceride levels were not reflected in changes in VOC profiles [11]. Technical factors that can contribute to variability can include instrument settings or scanning rate. GC separation is susceptible to minor RT variations during volatile physical separation; although, the use of TD technology provides a more consistent method for sample introduction into the column. The humidity of the FAIMS clean air supply can alter background noise and reduce the sensitivity. Additionally, perceptible but minor increases in scan rate were observed with current FAIMS settings. Robust deep learning approaches that can incorporate these effects should be evaluated when analyzing disease-associated volatolomic profiles. Data from raw detector outputs such as those used in this study are less amenable to noise filtering or other correction steps when compared with data generated from established chromatographic methods. Although many technical factors that can contribute to variation cannot be completely eliminated, their impact can be minimized by using meticulous collection and analytic protocols. The utility of volatolomic signatures as disease biomarkers will thus be highly dependent on disease-associated alterations that are of sufficiently greater magnitude to overcome some of these variations. As demonstrated in this pilot study, this is feasible for individuals with cirrhosis using GC-FAIMS.

Approaches using GC-MS-based VOC identification requires hands-on, stringent analysis by skilled personnel. Operator-generated discrepancies further increase the amount of non-biological information within the dataset. Automated assessment of raw instrument data output bypasses the need for manual specialist involvement and processing while ensuring consistency in detection and analysis. The supervised use of CNN trained on raw GC-MS abundance matrices based on time resolved mass-to-charge ratio has shown high sensitivity for VOC detection [12]. Automation of analysis would reduce the labor required for large cohort metabolomic studies and also provide a framework for standardization for multi-site studies that may enable detection of batch effects [13]. In addition, automated methods of assigning time or intensity defined descriptors to individual VOC verified through the use of standards could further result in the streamlined recognition of individual disease associated VOC.

Sample storage conditions are of particular importance for breath VOC analysis, but their effects can be mitigated by limiting the storage time of samples prior to analysis. Although storage of breath samples at 4˚C and analysis within 30 days has been recommended [14, 15], VOC stability and lack of storage artefacts were reported during storage for 1.5 months at -80˚C using dual-bed Tenax TA and Carbograph sorbent tubes. Our models performed best when trained and tested on samples that had not undergone prolonged storage. Sources of confounding artifacts during storage could result from the migration and separation of trapped VOC between beds within multi-sorbent tubes, leakage out of their caps, or contamination from VOC that diffuse into the storage tubes onto the sorbent material from the coolant, external environment, or from foreign substances adhering to the non-emitting tube caps.

The study has some limitations. The study cohort encompassed a broad range of diseases of diverse etiologies that may have variable metabolic effects on VOC production. Having demonstrated the feasibility within this context, further studies to determine the utility of volatolomic profiling as a biomarker of specific clinical phenotypes in disease-specific cohorts are warranted. A further limitation is the reliance on algorithmic approaches for data obtained from a single study site. The use of a novel separation approach precluded the validation in an independent setting. Cross-site validation studies will become possible once the approaches in this study have been adopted and implemented in other settings. These will require particular attention to evaluate for potential batch effects that could arise as a result of the collection or analysis environment, instrument use and operator practices. Standardization of volatolomic profiling across different sites will be necessary prior to further use as diagnostic biomarkers in practice. This would entail the development and use of volatolomic-centric quality control mixtures within and between studies to compare cross-study measurements and the use of within-batch correction algorithms to mitigate the impact of any batch effects that are observed [16].

The use of volatolomic signatures and machine learning to generate and analyze predictive biomarker profiles obviates the need for detailed identification of individual VOC. Future studies directed towards the targeted detection and identification of specific VOC metabolites that are informative components of the volatolomic biomarker profiles may be considered, and could eventually enhance our understanding of underlying disease pathophysiology.

## Supporting information

**S1 Fig. Data pre-processing pipeline.**
(PDF)

**S2 Fig. Separation based chromatograms from breath analysis of patients at different disease stages.**
(PDF)

**S3 Fig. Variability of molecular features with age or etiology.**
(PDF)

**S4 Fig. Inter-individual variability in separation chromatograms.**
(PDF)

**S5 Fig. Performance of classifier models in distinguishing across disease stages.**
(PDF)

## Acknowledgments

We are grateful for the contributions of all our study subjects and their participation, study coordination provided by Robert Brannock and Torsak Vimoktayon, CNN analysis provided by Matt Spiegel, and useful insights provided by members of the Patel lab.

## Author Contributions

**Conceptualization:** Tushar Patel.

**Data curation:** Jonathan N. Thomas, Tushar Patel.

**Formal analysis:** Jonathan N. Thomas, Tushar Patel.

**Funding acquisition:** Tushar Patel.

**Investigation:** Jonathan N. Thomas, Joanna Roopkumar, Tushar Patel.

**Methodology:** Jonathan N. Thomas, Tushar Patel.

**Project administration:** Tushar Patel.

**Resources:** Tushar Patel.

**Supervision:** Tushar Patel.

**Visualization:** Tushar Patel.

**Writing – original draft:** Jonathan N. Thomas, Tushar Patel.

**Writing – review & editing:** Jonathan N. Thomas, Tushar Patel.

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
