## [Decision Letter · Decision Letter 0]

27 May 2021

PONE-D-21-15285

Machine learning analysis of volatolomic profiles in breath can identify non-invasive biomarkers of liver disease: A pilot study

PLOS ONE

Dear Dr. Patel, Dear Tushar,

Thank you for submitting your manuscript to PLOS ONE. After careful consideration, we feel that it has merit but does not fully meet PLOS ONE’s publication criteria as it currently stands. Therefore, we invite you to submit a revised version of the manuscript that addresses the points raised during the review process.

We look forward to receiving your revised manuscript.

Kind regards,

Matias A Avila, Ph.D.

Academic Editor

PLOS ONE

Journal Requirements:

Reviewers' comments:

Reviewer's Responses to Questions

**Comments to the Author**

1. Is the manuscript technically sound, and do the data support the conclusions?

Reviewer #1: Yes

Reviewer #2: Yes

2. Has the statistical analysis been performed appropriately and rigorously? 

Reviewer #1: Yes

Reviewer #2: Yes

3. Have the authors made all data underlying the findings in their manuscript fully available?

Reviewer #1: Yes

Reviewer #2: Yes

4. Is the manuscript presented in an intelligible fashion and written in standard English?

Reviewer #1: Yes

Reviewer #2: Yes

5. Review Comments to the Author

Reviewer #1: The manuscript is technically sound and it represents an extensive piece of research in machine learning field.

I have only some reflections and considerations to share with the authors.

The major issues had been resolved adequately taking into account the limitation, principally the sample size. As far as it demonstrates the consistency of the data, techniques (5fold-CV, correctly defined train-validation cohorts) and the groups are well balanced, there is nothing to remark. In the case of the SVM and other mentioned approaches, when they reach a 100% sensitivity and the specificity falls, often is a result of sacrificing the smaller group classifying all samples into the bigger group, so the accuracy (normally the score used to fit the model) will reach the maximum.

The use of ensemble and tandem methods is particularly indicated when the data is complex and the individual markers have small to medium predictive capacity (AUC=0.5-0.7) as it is in this case. The implementation of ResNet50 is a foot in the door in the good sense, because it’s easily customizable to normalize and remove the noise from raw data, for this purpose is currently used in computer vision. So it can be easily seen the future application of it to automatize the analysis.

As a final remark, it will be very exciting to implement a reinforced learning to this approach, this will give to the algorithm the capacity of learning on each example and thereby it will be very useful in diagnostic units, but this is the future as far as it is a pilot study.

Reviewer #2: Thomas and colleagues demonstrate in this proof of concept study the feasibility and utility of breath volatolomic profiling to classify liver disease (cirrhosis or portal hypertension). The manuscript is well conducted, material and methods detailed and results well structured and supported.

I have only few minor suggestions:

1) Similar to determination of inter- and intra-individual variability across different samples or participant technical replicates, did authors evaluate the age-related variability? They have participants from 24 to 76 years old.

2) Did authors explore the correlation with the primary liver disease etiology?

3) In line 269, related to Figure 3 (graph on the right): authors have written that “four showed a relationship with disease stage”. I would suggest to explain that these four trend to increase with disease stage.

6. PLOS authors have the option to publish the peer review history of their article (what does this mean?). If published, this will include your full peer review and any attached files.

Reviewer #1: **Yes: **Jose Maria Herranz Alzueta

Reviewer #2: No

---

## [Author Response · Author response to Decision Letter 0]

22 Oct 2021

Response to reviewers.

We thank the editor and reviewers for their careful and thoughtful review of our manuscript and for their comments. We have responded to each of these below:

Reviewer #1: The manuscript is technically sound and it represents an extensive piece of research in machine learning field. I have only some reflections and considerations to share with the authors.

The major issues had been resolved adequately taking into account the limitation, principally the sample size. As far as it demonstrates the consistency of the data, techniques (5fold-CV, correctly defined train-validation cohorts) and the groups are well balanced, there is nothing to remark. In the case of the SVM and other mentioned approaches, when they reach a 100% sensitivity and the specificity falls, often is a result of sacrificing the smaller group classifying all samples into the bigger group, so the accuracy (normally the score used to fit the model) will reach the maximum. The use of ensemble and tandem methods is particularly indicated when the data is complex and the individual markers have small to medium predictive capacity (AUC=0.5-0.7) as it is in this case. The implementation of ResNet50 is a foot in the door in the good sense, because it’s easily customizable to normalize and remove the noise from raw data, for this purpose is currently used in computer vision. So it can be easily seen the future application of it to automatize the analysis. As a final remark, it will be very exciting to implement a reinforced learning to this approach, this will give to the algorithm the capacity of learning on each example and thereby it will be very useful in diagnostic units, but this is the future as far as it is a pilot study.

Response: We thank the reviewer for their helpful comments and insights. We concur that reinforced learning approaches could be interesting and we will embark on additional studies to explore these approaches in future.

Reviewer #2: Thomas and colleagues demonstrate in this proof of concept study the feasibility and utility of breath volatolomic profiling to classify liver disease (cirrhosis or portal hypertension). The manuscript is well conducted, material and methods detailed and results well-structured and supported. I have only few minor suggestions:

1) Similar to determination of inter- and intra-individual variability across different samples or participant technical replicates, did authors evaluate the age-related variability? They have participants from 24 to 76 years old.

Response: We have now examined the age-related variability in molecular features within biological replicates. A greater variability was observed in samples from participants aged 45yr or younger. These data are now included in the revised manuscript in lines 258-261, and the data included in the Supplementary Information as S3 Fig.

2) Did authors explore the correlation with the primary liver disease etiology?

Response: We have examined the average peak area, and the median SD of each molecular feature across replicates from individuals with disease of different etiology. These data are now included in the revised manuscript and the data included in the Supplementary Information as S3 Fig.

3) In line 269, related to Figure 3 (graph on the right): authors have written that “four showed a relationship with disease stage”. I would suggest to explain that these four trend to increase with disease stage.

Response: We thank the expert reviewer for their helpful comments and have revised this statement to clarify this as recommended by the reviewer.

---

## [Editor Report · Decision Letter 1]

3 Nov 2021

Machine learning analysis of volatolomic profiles in breath can identify non-invasive biomarkers of liver disease: A pilot study

PONE-D-21-15285R1

Dear Dr. Patel,

We’re pleased to inform you that your manuscript has been judged scientifically suitable for publication and will be formally accepted for publication once it meets all outstanding technical requirements.

Kind regards,

Matias A Avila, Ph.D.

Academic Editor

PLOS ONE
---

## [Editor Report · Acceptance letter]

17 Nov 2021

PONE-D-21-15285R1 

Machine learning analysis of volatolomic profiles in breath can identify non-invasive biomarkers of liver disease: A pilot study 

Dear Dr. Patel:

I'm pleased to inform you that your manuscript has been deemed suitable for publication in PLOS ONE. Congratulations! Your manuscript is now with our production department. 

Kind regards, 

on behalf of

Dr Matias A Avila 

Academic Editor

PLOS ONE